# CONTINUAL KNOWLEDGE GRAPH LINK PREDICTION: BEYOND EXPERIENCE REPLAY

## ABSTRACT

Knowledge graphs (KGs) empower AI systems with essential inference capabilities as they increasingly integrate into life and industries. The dynamic nature of real-world KGs underscores the necessity for KG link prediction methods to possess continual learning capabilities. However, the existing benchmark datasets primarily rely on sampling based methods, falling short of adequately evaluating models' abilities for continual KG link prediction. In this paper, we explicitly formulate the continual KG link prediction task and provide definitions for its two specific settings: class-incremental and expansive. Two new benchmark datasets are established to provide valid benchmarking for fair evaluation of continual KG link prediction methods. Furthermore, we propose BER, a novel approach based on experience replay and knowledge distillation to alleviate the catastrophic forgetting problem. Extensive experimental results demonstrate the datasets' effectiveness in providing a fair evaluation of continual learning ability and validate the efficacy of our proposed method. Codes can be found in supplementary material and will be released along with both datasets upon acceptance.

## 1 INTRODUCTION

Knowledge graphs (KGs) store real-world events as factual triplets denoted as $(h, r, t)$. Each triplet delineates a relation $r$ between a head entity $h$ and a tail entity $t$. The intrinsic incompleteness problem of KGs has expedited extensive research on KG link prediction (Lin et al., 2015), which aims to predict missing relations based on the existing knowledge contained in a KG.

As real-world KGs are often semi-automatically constructed and subsequently updated with newly emerging information, KGs inherently possess the dynamic nature of evolving over time. Typically, KGs are initially constructed with factual triplets associated with a group of relations. As time progresses, new factual triplets associated with these relations are added, along with the incorporation of new relations and their associated triplets. For instance, the relation "locatedIn" remains a constant presence in KGs but more new associated triplets will emerge. And new relations related to the Covid-19 pandemic, as well as their triplets, will appear from the year of 2019 onwards. Given these characteristics of KGs, it is both crucial and imperative for KG link prediction methods to incorporate the continual learning ability to constantly adapt to the newly emerging data.

Current KG link prediction methods typically assume that the same class space is shared during model training and testing stages, preventing them from incorporating new relations effectively. Training on newly emerging data will subsequently result in abrupt loss of previously acquired knowledge, a phenomenon known as catastrophic forgetting (French, 1999). Some recent studies resort to continual learning methods to accommodate the dynamic property of KGs and alleviate the catastrophic forgetting problem. For example, Daruna et al. (2021) introduces five continual learning inspired methods for continual KG embedding and propose a heuristic sampling strategy to uniformly sample triplets from a given KG to generate each continual learning task. TIE (Wu et al., 2021) proposes a framework centered on semantic KGs and generates yearly graph snapshots by transforming facts (timestamped triplets) with time intervals into multiple facts.

However, *there remains a noticeable absence of a clearly defined formulation for the continual KG link prediction task, coupled with a shortage of standardized benchmark datasets specifically designed for appropriately evaluating continual KG link prediction methods.* There are several limitations in previous works: (1) the heuristic sampling strategy proposed by (Daruna et al., 2021)

does not account for the actual timestamps of facts when dividing tasks, resulting in a mixture of facts from different time spans within a single task; (2) TIE has access to previous tasks numbered from $t-1$ to $t-5$ at a given time $t$, which violates the standard continual learning practice (Mirtaheri et al., 2023); (3) the dataset generation process in TIE (Wu et al., 2021) can result in an overlap of 95% between consecutive KG snapshots.

In this paper, we first give a formal definition for continual KG link prediction task and introduce a class-incremental benchmark dataset, named **ICEWS05-15-classIL**. To align with both the dynamics of KGs and the core tenets of continual learning, ICEWS05-15-classIL meets two key criteria: (1) To preserve the inherent temporal property of KGs, the task division should adhere to the actual timestamps associated with facts; (2) To effectively assess the models' continual learning capabilities in KG link prediction, the input data distribution should remain distinct across different tasks.

To emulate real-world scenarios where aside from the continuously emerging new relations, a certain group of relations tend to persist throughout the course of KG evolution, we derive a new **expansive continual learning setting** tailored to KGs. Accordingly, a new benchmark dataset, **ICEWS05-15-expansive**, is constructed to more faithfully replicate the real-world dynamics of KGs. Both datasets aim to provide a valid benchmarking for fairly evaluating the performance of continual learning methods on continual KG link prediction.

The aforementioned benchmarks uncover significant pitfalls of current continual learning techniques. To alleviate these problems, we propose a novel approach **BER** (beyond experience replay) for continual KG link prediction based on experience replay and knowledge distillation. Specifically, we devise an attention-based method to select representative facts for each relation and revisit those facts during the learning of a new task to alleviate catastrophic forgetting. In addition, we design an embedding distillation loss to compensate for the scarcity of replayed facts and enhance the memorization of previously learned knowledge. We also employ a contrastive learning based method to learn expressive entity/relation embeddings by modeling correlations between each fact and its context. BER exhibits superior performance compared to baseline methods, underscoring the significance of tailored methods for continual knowledge graph link prediction, as substantiated by our empirical analysis. Our contribution can be summarized as:

- To the best of our knowledge, we are the first to explicitly formulate the continual KG link prediction task and provide clear definitions for its two settings: class-incremental and expansive. The class-incremental setting fully aligns with the tenets of continual learning while the expansive setting is especially tailored to the real-world dynamics of KGs.
- We construct two benchmark datasets, ICEWS05-15-classIL and ICEWS05-15-expansive, to provide a valid benchmark for fair evaluation of continual KG link prediction methods.
- We propose BER for continual KG link prediction built upon experience replay and knowledge distillation, significantly mitigating the catastrophic forgetting problem. Extensive experiments on the two benchmark datasets validate the superiority of BER compared to other baseline methods.

## 2 RELATED WORK

### 2.1 TEMPORAL KG LINK PREDICTION

The current literature on temporal KG link prediction comprises two lines of methods. The first line focuses on enhancing KG link prediction through incorporating time-dependent representations (Dasgupta et al., 2018; Goel et al., 2020; Jiang et al., 2016; Xu et al., 2019). The second line employs spatial-temporal graph neural networks to model structural and temporal dependencies (Jin et al., 2019; Sankar et al., 2020). The objectives of temporal KG link prediction and continual KG link prediction are distinct: (1) the former utilizes temporal information to enhance the representation of timestamped factual triplets. However, they assume a static class space shared between training and testing phases. Consequently, they also face the issue of catastrophic forgetting when confronted with new streaming data. (2) Continual KG link prediction focuses on designing models to quickly incorporate newly emerging relations while not forgetting previously learned knowledge.

## 2.2 Continual Learning on Graphs

Continual learning, also known as lifelong learning or incremental learning, aims to learn a number of tasks sequentially while not forgetting previously acquired knowledge (French, 1999). Necessitated by the dynamic nature inherent in various forms of data, continual learning has been investigated in many areas such as computer vision (Aljundi et al., 2019a; Shin et al., 2017) and natural language processing (Biesialska et al., 2020; Sun et al., 2019). However, continual learning on graphs have been largely under-explored. To date, the existing approaches to continual learning on graphs can be classified into three categories: replay based methods, regularization based methods, and parametric isolation based methods.

**Replay based methods** revisit a collection of samples or generate samples of pseudo-data of the previously learned tasks during the learning of a new task to strike a balance between learning new information and retaining knowledge from prior tasks (Aljundi et al., 2019b; Caccia et al., 2020; Chrysakis & Moens, 2020; Knoblauch et al., 2020; Lopez-Paz & Ranzato, 2017; Shin et al., 2017). ER-GNN (Zhou & Cao, 2021) proposes three experience node selection strategies to pick representative nodes for replay. Ahrabian et al. (2021) proposes a structure-aware reservoir-based continual learning approach for recommender systems. **Regularization based methods** seek to maintain the model's performance on previous tasks using regularization terms to impose penalties on changes to the model's parameters (Farajtabar et al., 2020; Jung et al., 2016; Kirkpatrick et al., 2017; Li & Hoiem, 2017; Saha et al., 2020; Nguyen et al., 2018). Elastic weight consolidation (EWC) (Kirkpatrick et al., 2017) introduces a quadratic penalty to stabilize the model parameters and avoid excessive fluctuations. In continual graph learning, Topology-aware Weight Preserving (TWP) (Liu et al., 2021) explores the local graph structures and attempts to stabilize the key parameters in the topological aggregation to overcome catastrophic forgetting. **Parametric isolation based methods** introduce new parameters for new tasks so as to avoid severe modifications to parameters that capture key information from previous tasks (Rusu et al., 2016; Wortsman et al., 2020; Wu et al., 2019; Yoon et al., 2019; 2018). HPNs (Zhang et al., 2022) extract different levels of abstract information to represent the continuously expanded graphs.

Several recent works on continual KG link prediction include CKGE (Daruna et al., 2021) and TIE (Wu et al., 2021). Daruna et al. (2021) highlight the limitation in existing KG embedding algorithms, where they assume that all concepts are known *a priori* and require learning everything from scratch when new information arises. CKGE introduces the concept of continual knowledge graph embedding to address this limitation, and introduces a heuristic sampling strategy for generating CKGE datasets. TIE (Wu et al., 2021) combines knowledge graph representation learning, experience replay, and temporal regularization to improve model performance on temporal knowledge graph completion (TKGC) task. It introduces novel evaluation metrics to assess the ability of TKGC models to handle deleted facts and demonstrates the efficiency and effectiveness of the TIE framework through experiments on real-world datasets. However, we notice that the benchmark datasets constructed by CKGE and TIE fail to meet the criteria of continual KG link prediction, *i.e.* real timestamps of facts are neglected and/or the overlap of consecutive KG snapshots.

## 3 Problem Formulation

In this section, we formally define continual KG link prediction and its two settings. The notations used in this paper are summarized in Appendix A.

**Definition 1** *Continual KG Link Prediction*. Given a temporal KG represented as $\mathcal{G} = \{\mathcal{E}, \mathcal{R}, \mathcal{TS}, \mathcal{F}\}$, where $\mathcal{E}, \mathcal{R}, \mathcal{TS}$ and $\mathcal{F}$ denote the sets of entities, relations, timestamps and facts respectively, continual KG link prediction considers a sequence of tasks, denoted as $\{\mathcal{T}_1, \mathcal{T}_2, ..., \mathcal{T}_T\}$, with $\mathcal{T}_i$ consists of a snapshot of $\mathcal{G}$, denoted as $\mathcal{G}_i = \{\mathcal{E}_i, \mathcal{R}_i, \mathcal{TS}_i, \mathcal{F}_i\}$. The $i$-th task has its own dataset $\mathcal{D}_i$, which is a subset of $\mathcal{F}_i$. $\mathcal{D}_i$ is further divided into non-overlapping training, validation and testing sets, denoted as $\mathcal{D}_i^{train}, \mathcal{D}_i^{val}$ and $\mathcal{D}_i^{test}$, respectively. In $\mathcal{T}_i$, a continual KG link prediction method seeks to learn from $\mathcal{D}_i^{train}$ to incorporate the newly emerging relations $\mathcal{R}_i$ and exploit the existing facts in a KG to predict missing ones. This boils down to predicting the correct entity that completes $(h, r, ?)$ or $(?, r, t)$ for a given relation $r$. The evaluation is performed on $\tilde{\mathcal{D}}_i^{val} = \bigcup_{j=1}^i \mathcal{D}_j^{val}$ to assess the model's ability of continuously learning new knowledge while preserving the knowledge learned from previous tasks.

In the following, we first provide a formal definition for the standard class-incremental (class-IL) setting (Van de Ven & Tolias, 2019) for continual KG link prediction. In the real world, KGs often involve specific relations that persist throughout the dynamic evolution, yet new facts associated with these relations continue to emerge. To better emulate this condition, we derive a new **expansive** setting tailored to KGs. This setting accommodates the presence of a set of persistent relations across tasks, aligning more closely with the complexities of real-world KGs.

**Definition 2** *Class-IL Continual KG Link Prediction*. Given a sequence of tasks $\{\mathcal{T}_1, \mathcal{T}_2, ..., \mathcal{T}_T\}$, where $\mathcal{T}_i$ consists of a snapshot KG, denoted as $\mathcal{G}_i = \{\mathcal{E}_i, \mathcal{R}_i, \mathcal{TS}_i, \mathcal{F}_i\}$. The collection of relations in $\mathcal{T}_i$ is denoted as $\mathcal{R}_{\mathcal{T}_i}$. Under class-IL setting, for any distinct tasks $\mathcal{T}_m$ and $\mathcal{T}_n$, the input data distributions are not equivalent, holding true for all cases where $m$ is not equal to $n$. That is, $\mathcal{R}_{\mathcal{T}_i} \cap \mathcal{R}_{\mathcal{T}_j} = \emptyset$, holding true for all cases where $i$ is not equal to $j$. A class-IL continual KG link prediction method aims to undertake continual KG link prediction as predefined in **Definition 1** under the aforementioned constraints.

**Definition 3** *Expansive Continual KG Link Prediction*. Given a sequence of tasks $\{\mathcal{T}_1, \mathcal{T}_2, ..., \mathcal{T}_T\}$, where $\mathcal{T}_i$ consists of a snapshot KG, denoted as $\mathcal{G}_i = \{\mathcal{E}_i, \mathcal{R}_i, \mathcal{TS}_i, \mathcal{F}_i\}$. Under expansive setting, each $\mathcal{T}_i$ contains two types of relations: (1) persistent relations that remain constant presence across tasks, denoted as $\mathcal{R}_p$; (2) new relations that are introduced by new tasks, denoted as $\mathcal{R}_{\mathcal{T}_i}$. The arrival of new task $\mathcal{T}_{i+1}$ results in the introduction of new facts associated with $\mathcal{R}_p$, and new relations $\mathcal{R}_{\mathcal{T}_{i+1}}$ and their associated facts, making expansive setting better replicate the nature of KGs. An expansive continual KG link prediction method aims to undertake continual KG link prediction as predefined in **Definition 1** under the aforementioned constraints.

## 4 DATASET CONSTRUCTION

In the current continual KG link prediction literature, existing benchmark datasets such as WN18RR-5-LS and FB15K237-5-LS (Daruna et al., 2021) do not guarantee the distinct class space across different tasks. These datasets consider a same group of relations in different tasks, which violates the original intention of continual learning. The dataset generated by (Wu et al., 2021) also suffers from an overlap of 95% between consecutive KG snapshots (Mirtaheri et al., 2023). Furthermore, current benchmark datasets do not incorporate the actual timestamps of facts as an intrinsic and informative basis for task division. To facilitate the future research on continual KG link prediction, we construct a benchmark dataset under class-IL setting based on ICEWS05-15 (Garcia-Duran et al., 2018) (Sec 4.1) and further propose a new expansive setting (Sec 4.2), which is more in line with the characteristics of KGs and more practical for real-world scenarios.

The Integrated Crisis Early Warning System dataset, often referred to as ICEWS05-15 (Garcia-Duran et al., 2018) is a collection of event data, covering a time span from 2005 to 2015. Each event in ICEWS05-15 is structured and comes with a specific timestamp, accurately recording the evolving situation over time. In the following, we describe two new benchmark datasets for continual KG link prediction, ICEWS05-15-classIL and ICEWS05-15-expansive, both derived from ICEWS05-15. The detailed statistics of the two datasets are listed in Table. 1.

### 4.1 ICEWS05-15-CLASSIL

We partition the ICEWS05-15 dataset into six sequential tasks, $\{\mathcal{T}_1, \mathcal{T}_2, ..., \mathcal{T}_6\}$, based on their respective time spans $\mathcal{TS}_i$. To meet the first criterion of preserving the temporal property, we construct $\mathcal{D}_i$ by selecting facts associated with $\mathcal{R}_{\mathcal{T}_i}$ and timestamped within $\mathcal{TS}_i$, i.e. $\mathcal{D}_i = \{(h, r, t, ts)|r \in \mathcal{R}_{\mathcal{T}_i}, ts \in \mathcal{TS}_i\}$. To ensure distinct input distributions for different tasks, three distinct relations $\mathcal{R}_{\mathcal{T}_i} = \{\mathcal{R}_{\mathcal{T}_{i1}}, \mathcal{R}_{\mathcal{T}_{i2}}, \mathcal{R}_{\mathcal{T}_{i3}}\}$ are selected for each task $\mathcal{T}_i$ based on their frequency to serve as the target relations, adhering to the second criterion of distinct input data distribution.

In doing so, each task $\mathcal{T}_i$ contains a distinct snapshot composed of facts originated from the same time interval $\mathcal{TS}_i$ while shares no overlap of relations, ensuring that $\mathcal{R}_{\mathcal{T}_i} \cap \mathcal{R}_{\mathcal{T}_j} = \emptyset$ and $\mathcal{TS}_i \cap \mathcal{TS}_j = \emptyset$, holding true for all cases where $i$ is not equal to $j$. Specifically, all facts in $\mathcal{T}_1$ are timestamped from January 2005 to October 2006 and all facts in $\mathcal{T}_2$ are timestamped from November 2006 to August 2008, *etc*. And the selection criteria for target relations is that each selected $\mathcal{R}_{\mathcal{T}_{ij}}(1 \leq j \leq 3)$ is associated with less than 10,000 but more than 1,000 facts in $\mathcal{F}_i$. To maintain the semantic connections and knowledge inference ability of KGs, we also assume that the method

| Dataset | Statistic | Task1 | Task2 | Task3 | Task4 | Task5 | Task6 |
|---|---|---|---|---|---|---|---|
| ICEWS05-15-classIL | $\|\mathcal{E}_i\|$ | 1,654 | 1,241 | 1,308 | 1,775 | 1,693 | 1,624 |
| | $\|\mathcal{R}_i\| \,/\, \|\mathcal{R}_p\|$ | 3 / 0 | 3 / 0 | 3 / 0 | 3 / 0 | 3 / 0 | 3 / 0 |
| | $\|\mathcal{F}_i\|$ | 8,538 | 6,920 | 7,146 | 11,043 | 7,249 | 8,728 |
| ICEWS05-15-expansive | $\|\mathcal{E}_i\|$ | 3,622 | 3,575 | 3,306 | 3,267 | 3,323 | 3,559 |
| | $\|\mathcal{R}_i\| \,/\, \|\mathcal{R}_p\|$ | 8 / 5 | 8 / 5 | 8 / 5 | 8 / 5 | 8 / 5 | 8 / 5 |
| | $\|\mathcal{F}_i\|$ | 25,096 | 24,483 | 23,647 | 24,706 | 20,963 | 25,116 |

Table 1: Number of facts contained in each task. $\|\mathcal{E}_i\|$ is the number of entities, $\|\mathcal{R}_i\|$ is the number of relations, $\|\mathcal{R}_p\|$ is the number of overlapped relations in all tasks, and $\|\mathcal{F}_i\|$ is the number of facts.

| Dataset | $\|\mathcal{E}\|$ | $\|\mathcal{R}\|$ | $\|\mathcal{D}^{train}\|$ | $\|\mathcal{D}^{val}\|$ | $\|\mathcal{D}^{test}\|$ | $\|\mathcal{T}\|$ | Division Basis | Overlap |
|---|---|---|---|---|---|---|---|---|
| WN18RR-5-LS | 20,471 | 11 | 86,835 | 5,819 | 5,893 | 5 | Random | 100% |
| FB15K237-5-LS | 13,163 | 237 | 272,115 | 84,646 | 98,796 | 5 | Random | 100% |
| YAGO11k-TIE | 10,623 | 10 | 215,894 | 23,197 | 22,567 | 61 | Timestamp | 95% |
| Wikidata12k-TIE | 12,554 | 24 | 257,542 | 20,764 | 19,746 | 78 | Timestamp | 95% |
| **ICEWS05-15-classIL** | 4,511 | 18 | 34,736 | 9,925 | 4,963 | 6 | Timestamp | 0% |
| **ICEWS05-15-expansive** | 7,968 | 23 | 100,807 | 28,802 | 14,401 | 6 | Timestamp | 22% |

Table 2: Comparison of related benchmark datasets. $\|\mathcal{E}\|$, $\|\mathcal{R}\|$ and $\|\mathcal{T}\|$ indicate the number of the entities, relations and tasks in corresponding dataset. $\|\mathcal{D}^{train}\|$, $\|\mathcal{D}^{val}\|$ and $\|\mathcal{D}^{test}\|$ indicate the number of facts in training set, validation set and testing set, respectively.

has access to a background knowledge graph $\mathcal{G}'_i$ in $\mathcal{T}_i$, which is a subset of $\mathcal{G}_i$ with all the facts associated $\mathcal{R}_{\mathcal{T}_{ij}}(1 \leq j \leq 3)$ removed.

## 4.2 ICEWS05-15-EXPANSIVE

ICESW05-15-expansive is constructed by accommodating a group of persistent relations $\mathcal{R}_p = \{\mathcal{R}_{p1}, ..., \mathcal{R}_{p5}\}$ to each task that persist over the evolve of KGs. Similarly, facts that are associated with $\mathcal{R}_p$ and also timestamped within $\mathcal{TS}_i$ are added to $\mathcal{D}_i$. The selection criteria for persistent relations is that the number of facts associated with $\mathcal{R}_{pj}(1 \leq j \leq 5)$ is more than 500 in $\mathcal{F}_i(1 \leq i \leq 6)$, excluding $\mathcal{R}_{\mathcal{T}_i}$. A back ground graph $\mathcal{G}'_i$ is also constructed. The inclusion of persistent relations introduces a 22% overlap of relations across different tasks but brings the continual KG link prediction task into closer alignment with the real-world dynamics of KGs.

## 4.3 COMPARISON WITH PREVIOUS DATASETS

Table. 2 illustrates the comparison between the proposed ICEWS05-15-classIL, ICEWS05-15-expansive and other related benchmark datasets. Compared with WN18RR-5-LS and FB15K237-5-LS (Daruna et al., 2021), ICEWS05-15-classIL and ICEWS05-15-expansive employ timestamps as the basis for task division, ensuring that all facts within one task originate from the same time span. The partitioning approach of ICEWS05-15-classIL additionally guarantees a 0% overlap of relations across distinct tasks, in contrast to 100% overlap observed in WN18RR-5-LS and FB15K237-5-LS, and a 95% overlap in YAGO11k-TIE and Wikidata12k-TIE (Wu et al., 2021), establishing ICEWS05-15-classIL a standard continual learning dataset for KG link prediction task. The inclusion of $\mathcal{R}_p$ results in a 22% overlap in the relaxed version, ICEWS05-15-expansive, reducing the challenge but better replicate the real-world dynamics of KGs that a certain group of relations remain constant presence in KGs.

## 5 THE PROPOSED METHOD

We propose a new continual KG link prediction method **BER** based on experience replay and knowledge distillation. There are two key components in BER enhancing the model's continual learning ability: attention-based experience replay and embedding distillation. As BER is agnostic to KG link prediction methods, we give detailed description to a TransE-instantiated BER in this section.

## 5.1 TransE-based KG Link Prediction

In each task $\mathcal{T}_i$, we leverage the scoring function inspired by TransE (Bordes et al., 2013) which assumes the translational relationship holds between the embeddings of the head entity and the tail entity, *i.e.* $\mathbf{h} + \mathbf{r} = \mathbf{t}$. The scoring function is then defined as:

$$s_{(h_i, t_i)} = ||\mathbf{h}_i + \mathbf{r} - \mathbf{t}_i||_2, \tag{1}$$

where $||\mathbf{x}||_2$ denotes the $\ell_2$ norm of vector $\mathbf{x}$. With the scoring function, we obtain the following translational loss,

$$\mathcal{L}_t = \Sigma_{(h_i, t_i) \in \mathcal{D}_i^{train}} [\gamma + s_{(h_i, t_i)} - s_{(h_i, t_i')}]_+, \tag{2}$$

where $[x]_+$ represents the positive part of $x$. $s_{(h_i, t_i')}$ calculates the score of a negative pair $(h_i, t_i')$, synthesized by negative sampling of the positive pair $(h_i, t_i) \in \mathcal{D}_i^{train}$, *i.e.* $(h_i, r, t_i') \notin \mathcal{D}_i^{train}$. $\gamma$ is a hyper-parameter determining the margin to separate positive pairs from negative pairs.

## 5.2 Attention-based Experience Replay

### 5.2.1 Replay Fact Selection

Given a task $\mathcal{T}_i$ containing facts associated with relations from $\mathcal{R}_{\mathcal{T}_i}$, we aim to extract a selected group of representative facts for each relation $r \in \mathcal{R}_{\mathcal{T}_i}$. These representative facts are stored and replayed when new tasks come in, so as to

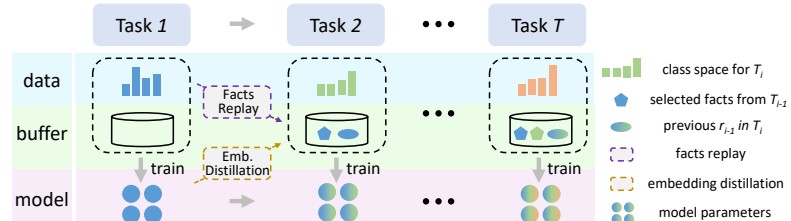

Figure 1: General framework of BER.

efficiently retain the knowledge learned from $\mathcal{T}_i$. The main consideration in our model design is that those representative facts should epitomize the relation $r$ they are associated with. Therefore, we design our fact selection process based on attention mechanism and select those facts that have the strongest similarity with other triplets associated with $r$.

Mathematically, given a batch of facts $\{(h_i, r_i, t_i, ts_i) | (h_i, r_i, t_i, ts_i) \in \mathcal{D}_i^{train}, r_i \in \mathcal{R}_{\mathcal{T}_i}\}$, each fact is first encoded as $\mathbf{x}_i = \mathbf{h}_i \oplus \mathbf{t}_i$, where $\mathbf{h}_i \in \mathbb{R}^d$ and $\mathbf{t}_i \in \mathbb{R}^d$ are the embeddings of entity $\mathbf{h}_i$ and tail entity $\mathbf{t}_i$ with dimension $d$, and $\mathbf{x} \oplus \mathbf{y}$ indicates the concatenation of two vectors $\mathbf{x}$ and $\mathbf{y}$. For all facts associated with the same relation $r_i$, our attention-based fact selection module aims to compare the similarity among facts and select those with the highest similarity with all other facts. To achieve this goal, we leverage a self-attention module on all facts associated with $r_i$:

$$\mathbf{X} = [\mathbf{x}_0; \mathbf{x}_1; ..; \mathbf{x}_n], \quad \mathbf{x}_i \in \mathbb{R}^{2d}, \quad 0 \le i \le n, \tag{3}$$

This input $\mathbf{X}$ is first transformed into two different matrices: the query matrix $\mathbf{Q} \in \mathbb{R}^{n \times d_v}$ and the key matrix $\mathbf{K} \in \mathbb{R}^{n \times d_k}$ with dimension $d_q = d_k = d_c$. The self-attention matrix is then computed using $\mathbf{Q}$ and $\mathbf{K}$.

$$\text{Attention}(\mathbf{Q}, \mathbf{K}) = \text{softmax}(\mathbf{Q} \cdot \mathbf{K}^\top / \sqrt{d_k}), \tag{4}$$

where $\text{Attention}_{ij}$ denotes the similarity score of $\mathbf{x}_i$ and $\mathbf{x}_j$. Based on the self-attention matrix, the attention rank score of $\mathbf{x}_i$ can be computed by:

$$\text{AttnRank}_i = \sum_{j=0}^{n} \text{Attention}_{ij}, \tag{5}$$

The attention rank score can describe how similar each fact is to all other facts associated with the same relation. We select the top $N$ facts based on its $\text{AttnRank}$ from each batch and union all the facts selected from different batches in one epoch $j$ to generate the fact set $\mathcal{FS}_{r_i}^j$ for relation $r_i$. The final representative fact set for $r_i$, denoted as $\mathcal{FS}_{r_i}$, is generated by:

$$\mathcal{FS}_{r_i} = \bigcap_{j=0}^{E} \mathcal{FS}_{r_i}^j, \tag{6}$$

where $E$ denotes the number of epochs in the current task.

### 5.2.2 CONTRASTIVE LEARNING BASED REPRESENTATION REFINEMENT

The representativeness of selected facts for replay is closely related to the expressiveness of entity embeddings. Therefore, we leverage a contrastive learning based representation refinement inspired by HiRe (Wu et al., 2023) to reinforce the embedding learning for each fact. For a given a fact $(h, r, t, ts)$, we denote its wider context as $\mathcal{C}_{(h,r,t,ts)} = \mathcal{N}_h \cup \mathcal{N}_t$, where $\mathcal{N}_h = \{(r_j, t_j) | (h, r_j, t_j, ts_j) \in \mathcal{F}_i\}$ and $\mathcal{N}_t = \{(r_j, t_j) | (t, r_j, t_j, ts_j) \in \mathcal{F}_i\}$. Each relation-entity tuple $(r_j, t_j) \in \mathcal{C}_{(h,r,t,ts)}$ is first encoded as $\mathbf{re}_j = \mathbf{r}_j \oplus \mathbf{t}_j$, where $\mathbf{r}_j \in \mathbb{R}^d$ and $\mathbf{t}_j \in \mathbb{R}^d$ are the relation and entity embedding, respectively. A multi-head self-attention (MSA) block is then employed to uncover the underlying relationships within the context and generate context embedding $\mathbf{c}$:

$$\mathbf{c}_0 = [\mathbf{re}_1; \mathbf{re}_2; ...; \mathbf{re}_K], \qquad K = |\mathcal{C}_{(h,r,t,ts)}|, \tag{7}$$

$$\mathbf{c} = \sum_{j=0}^{K} \alpha \cdot \mathbf{re}_j, \qquad \alpha = \text{MSA}(\mathbf{c}_0), \tag{8}$$

where $\mathbf{c}_0$ is the concatenation of the embeddings of all relation-entity tuples and $|x|$ is the size of set $x$. The self-attention scores among all relation-entity tuples from $\mathcal{C}_{(h,r,t,ts)}$ can be computed by Eq. 8. A group of false contexts $\{\tilde{\mathcal{C}}_{(h,r,t,ts)j}\}$ are also synthesized by randomly corrupting the relation or entity of each relation-entity tuple $(r_j, t_j) \in \mathcal{C}_{(h,r,t,ts)}$. The embedding of each false context $\tilde{\mathcal{C}}_{(h,r,t,ts)j}$ can be learned via the context encoder as $\tilde{\mathbf{c}}_i$. Then, we use a contrastive loss to pull close the embedding of the target fact with its true context and to push away from its false contexts. The contrastive loss function is defined as follows:

$$\mathcal{L}_c = -\log \frac{\exp(\text{sim}(\mathbf{h} \oplus \mathbf{t}, \mathbf{c})/\tau)}{\sum_{j=0}^{N} \exp(\text{sim}(\mathbf{h} \oplus \mathbf{t}, \tilde{\mathbf{c}}_j)/\tau)}, \tag{9}$$

where $\mathbf{h}$ and $\mathbf{t}$ are the embeddings of entity $h$ and $t$, $N$ is the number of false contexts for $(h, r, t, ts)$, $\tau$ denotes the temperature parameter, $\text{sim}(x, y)$ measures the cosine similarity between $x$ and $y$.

### 5.3 EMBEDDING DISTILLATION

The challenge of continual KG link prediction lies in the catastrophic forgetting problem when new tasks come in. In addition to the replay of selected representative fact set $\mathcal{FS}_{r_i}$ in new tasks to alleviate the problem, we also propose an embedding distillation loss, aim to consolidate the memorization of previous knowledge by reviewing previously learned relation embeddings.

Considering a new task $\mathcal{T}_{i+1}$, the replay of $\mathcal{FS}_{r_i}$ results in the re-learning of $r_i$ and update $\mathbf{r}_i$ as $\mathbf{r}'_i$. However, the model is inclined toward $\mathcal{T}_{i+1}$ and the newly computed $\mathbf{r}'_i$ is subpar compared with $\mathbf{r}_i$, due to the limited number of replayed facts contained in $\mathcal{FS}_{r_i}$. To address this problem, we strengthen the model's capacity to retain previously acquired knowledge by imposing constraints on the update process for relation embeddings:

$$\mathcal{L}_{ED} = ||\mathbf{r}_i - \mathbf{r}'_i||_2^2, \tag{10}$$

The overall loss function of **BER** in $\mathcal{T}_i, i \geq 1$ is:

$$\mathcal{L} = \mathcal{L}_t + \beta_0 \mathcal{L}_c + \mathbb{1}(i > 1)\beta_1 \mathcal{L}_{ED}, \tag{11}$$

where $\mathbb{1}(\cdot)$ is an indicator function.

## 6 EXPERIMENTS

### 6.1 BASELINES AND EVALUATION METRICS

We conduct experiments on ICEWS05-15-classIL and ICEWS05-15-expansive. The detailed statistics of two datasets are provided in Section 4. We evaluate the performance using both MRR (mean reciprocal rank of correct entities) and Hits@10 (the proportion of correct entities ranked within the top 10). For fair comparison, we follow CKGE (Daruna et al., 2021) and compare the proposed BER against representative continual learning methods: PNN (Rusu et al., 2016), CWR (Lomonaco & Maltoni, 2017), L2 (Kirkpatrick et al., 2017). We also fine-tune the model with examples only

**ICEWS05-15-classIL**

| Methods | Task1 | | Task2 | | Task3 | | Task4 | | Task5 | | Task6 | | Average | |
|---|---|---|---|---|---|---|---|---|---|---|---|---|---|---|
| | MRR | Hits@10 | MRR | Hits@10 | MRR | Hits@10 | MRR | Hits@10 | MRR | Hits@10 | MRR | Hits@10 | MRR | Hits@10 |
| PNN | 29.06 | 50.18 | 10.82 | 26.23 | 5.69 | 17.34 | 5.10 | 17.12 | 6.97 | 20.62 | 6.47 | 20.93 | 10.69 | 25.40 |
| CWR | 13.91 | 33.16 | 27.16 | 46.24 | 12.08 | 30.14 | 11.38 | 28.80 | 13.21 | 35.38 | 22.96 | 46.85 | 16.78 | 36.76 |
| FT | 10.45 | 28.92 | 32.36 | 58.60 | 11.43 | 27.03 | 11.60 | 33.00 | 21.08 | 40.93 | 18.21 | 38.35 | 17.52 | 37.81 |
| L2 | 8.77 | 21.87 | 28.61 | 47.40 | 10.50 | 26.71 | 15.98 | 41.67 | 21.65 | 49.17 | 22.24 | 50.92 | 17.96 | 39.62 |
| **Ours** | 11.25 | 32.90 | 42.28 | 65.63 | 16.93 | 38.06 | 12.18 | 34.24 | 25.81 | 53.04 | 23.66 | 52.04 | **22.02** | **45.99** |

**ICEWS05-15-expansive**

| Methods | Task1 | | Task2 | | Task3 | | Task4 | | Task5 | | Task6 | | Average | |
|---|---|---|---|---|---|---|---|---|---|---|---|---|---|---|
| | MRR | Hits@10 | MRR | Hits@10 | MRR | Hits@10 | MRR | Hits@10 | MRR | Hits@10 | MRR | Hits@10 | MRR | Hits@10 |
| PNN | 35.96 | 52.17 | 28.78 | 42.28 | 20.72 | 31.42 | 18.02 | 28.84 | 20.09 | 32.08 | 17.90 | 27.96 | 23.58 | 35.79 |
| CWR | 14.68 | 32.60 | 20.26 | 39.52 | 14.81 | 32.60 | 14.19 | 32.80 | 16.74 | 36.65 | 34.10 | 51.72 | 19.13 | 37.65 |
| FT | 17.88 | 37.94 | 26.51 | 50.28 | 21.51 | 43.70 | 19.35 | 39.91 | 26.39 | 50.58 | 24.68 | 49.50 | 22.72 | 45.32 |
| L2 | 19.82 | 38.34 | 29.84 | 51.02 | 20.24 | 39.90 | 20.87 | 40.63 | 28.16 | 49.26 | 27.12 | 46.06 | 24.34 | 44.20 |
| **Ours** | 21.91 | 44.71 | 31.05 | 59.55 | 24.28 | 49.93 | 24.56 | 49.15 | 30.11 | 57.34 | 26.86 | 55.01 | **26.46** | **52.62** |

Table 3: Comparison against baseline methods on ICEWS05-15-classIL and ICEWS05-15-expansive.The results listed here indicate the final performance after training on Task 6.

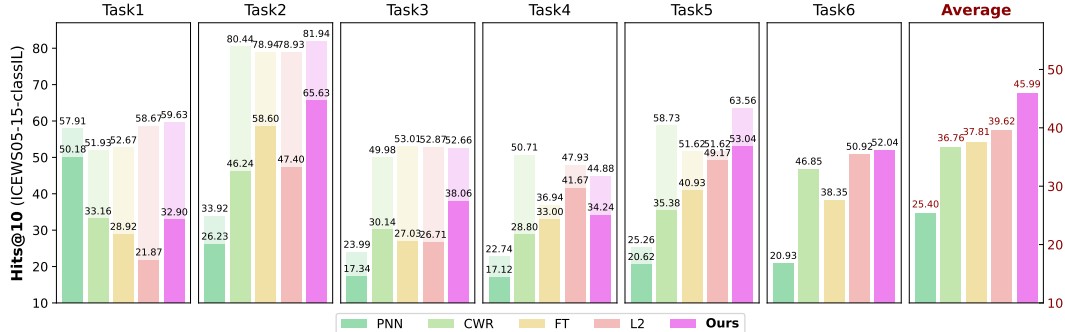

Figure 2: Comparison between different methods of Hits@10 on ICEWS05-15-classIL (results of MRR are in Appendix. C). We present the results spanning from Task 1 to Task 6. The lighter ones correspond to the testing performance immediately after training on each respective task, while the darker results indicate the final performance after training on Task 6 (with some amount of forgetting excluding Task 6).

from the current task, denoted as FT. Due to the fact that TIE (Wu et al., 2021) has access to the past 5 tasks which does not align with our settings, we leave TIE out in our comparison. All reported results are produced under the same experimental setting. All models are implemented in PyTorch and trained on a single V100 GPU. The detailed experimental settings are provided in the Appendix B.

## 6.2 COMPARISON WITH BASELINE METHODS

Table. 3 compares the performance of BER on each task against baseline methods after training on all six tasks on ICEWS05-15-classIL and ICEWS05-15-expansive, respectively. In general, BER achieves the best overall performance in terms of MRR and Hits@10 on both datasets and strikes a balance between learning new knowledge and maintaining knowledge learned from previous tasks, which validates its efficacy for continual KG link prediction task. Specifically, PNN tackles with catastrophic forgetting by freezing existing weights when new tasks come in and adding copies of existing layers for new tasks. This explains the reason why PNN achieving satisfactory results only on $\mathcal{T}_1$ but failing to effectively learn subsequent tasks. As for performance gains in terms of average MRR and Hits@10, BER surpasses the second best performer by +4.05% and +6.33% on ICEWS05-15-classIL, and by +2.12% and +7.3% on ICEWS05-15-expansive. BER achieves large performance improvements in terms of both metrics, proving the necessity of tailored method for continual KG link prediction task.

To further compare the ability of different methods to mitigate the catastrophic forgetting problem, Figure. 2 illustrates the comparison between testing performance obtained immediately after train-

| Ablation on ↓ | Components | | | Avg Metric | |
|---|---|---|---|---|---|
| | Rep. | Con. | Dis. | MRR | Hits@10 |
| BER (ours) | ✓ | ✓ | ✓ | 22.02 | 45.99 |
| w/o replay | ✗ | ✓ | ✓ | 19.34 | 41.61 |
| w/o contrastive | ✓ | ✗ | ✓ | 20.99 | 43.11 |
| w/o distillation | ✓ | ✓ | ✗ | 19.93 | 42.08 |

(a) Impact of different components.

| Number | Rep. | Ratio of facts | MRR | Hits@10 |
|---|---|---|---|---|
| 0 | ✗ | 0.0% | 19.34 | 41.61 |
| 4 | ✓ | 4.5% | 20.95 | 44.50 |
| 8 | ✓ | 9.2% | 21.68 | 45.31 |
| 10 | ✓ | 12.0% | 22.02 | 45.99 |
| 12 | ✓ | 16.7% | 22.26 | 46.34 |

(b) Impact of replay number.

| Number | Random based | | Attention based | |
|---|---|---|---|---|
| | MRR | Hits@10 | MRR | Hits@10 |
| 4 | 20.21 | 42.67 | 20.95 | 44.50 |
| 10 | 21.04 | 43.75 | 22.02 | 45.99 |

(c) Impact of manners in selecting replay fact.

| $\beta_0$ | 0 | 5 | 10 | 20 | 30 | 40 |
|---|---|---|---|---|---|---|
| MRR | 20.99 | 21.37 | 21.40 | 22.02 | 21.88 | 21.56 |
| Hits@10 | 43.11 | 44.97 | 44.68 | 45.99 | 45.63 | 45.02 |

(d) Coefficient of contrastive loss.

| $\beta_1$ | 0 | 5 | 10 | 20 | 30 | 40 |
|---|---|---|---|---|---|---|
| MRR | 19.93 | 21.04 | 21.75 | 21.77 | 22.02 | 21.68 |
| Hits@10 | 42.08 | 43.56 | 45.03 | 45.70 | 45.99 | 45.89 |

(e) Coefficient of distillation loss.

Table 4: Ablation studies on several key components in ICEWS05-15-classIL. We study: a) different components; b) different number of top $N$; c) different manners in selecting replay facts; d) coefficient of contrastive loss $\beta_0$; e) coefficient of review loss $\beta_1$; Default settings are marked in  gray .

ing on each respective task, and the final performance obtained after training on the last Task 6. The difference between light and dark colors indicate the degree of catastrophic forgetting. Our proposed BER demonstrates significantly lower levels of forgetting previously learned knowledge while effectively learning new tasks compared to other baseline methods.

### 6.3 ABLATION STUDY AND HYPER-PARAMETER SENSITIVITY

Our proposed continual KG link prediction method BER consists of three key components. To investigate the contributions of each component to the overall performance, we conduct a thorough ablation study on ICEWS05-15-classIL, as shown in Table 4. **Facts replay**: The replay of representative facts proves beneficial in retaining learned knowledge during continual learning. When the replay mechanism is removed from BER, the Hits@10 performance experiences a significant drop from 46.0% to 41.6%. Table 4(c) ablates different manners in selecting replay facts. The corresponding results highlight the importance of our attention-based fact selection approach in mitigating catastrophic forgetting in the continual KG link prediction task. Furthermore, we investigate the impact of the number of selected facts, as shown in Table 4(b). As expected, higher selection proportions yield better results but also increase the storage burden on the buffer. We find that a fact proportion of approximately 12% strikes a favorable trade-off between cost and performance. **Contrastive loss**: We experiment with different coefficients of contrastive loss in Eq. 11 to investigate the impact of representation refinement. Setting $\beta_0$ to 0 implies the absence of representation refinement in our BER model. We observe that representation refinement leads to a 1% improvement in MRR and a nearly 3% enhancement in Hits@10. This indicates the importance of leveraging the context of each fact in our approach. **Embedding Distillation**: We further investigate the effectiveness of embedding distillation by experimenting with different coefficients of $\mathcal{L}_{ED}$ in Eq. 11. $\beta_1 = 0$ means that embedding distillation is not incorporated in BER. Interestingly, we observe that BER performs well across a range of $\beta_1$ values from 10 to 40, highlighting the robustness and importance of embedding distillation in consolidating the memorization of previous knowledge.

## 7 CONCLUSION

This paper formally defines the class-incremental and expansive setting of continual KG link prediction task, and establishes benchmark datasets, ICEWS05-15-classIL and ICEWS05-15-expansive, for each setting. The proposed datasets are designed to support research on the dynamic characteristics of knowledge graphs (KGs) and enhance the continual learning capabilities of KG link prediction methods. We provide comprehensive use cases for these datasets, including the benchmarking of various continual learning methods for continual KG link prediction. We also propose BER, a novel continual KG link prediction method based on experience replay and knowledge distillation tailored for continual KG link prediction. Experimental results on the two benchmark datasets show that BER consistently alleviate the catastrophic forgetting problem. The ablation study and hyper-parameter study also verify the efficacy of key components in BER.

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

## A  Notations

The notations and symbols used in this paper are summarized in Table 5.

Table 5: Notations and Symbols.

| Symbol | Description |
|---|---|
| $\mathcal{G}$ | Knowledge graph. |
| $\mathcal{E}, \mathcal{R}, \mathcal{TS}, \mathcal{F}$ | Entity, relation, timestamp and fact set of a temporal KG. |
| $h, t$ | Head entity, tail entity. |
| $r$ | Relation. |
| $ts$ | Timestamp. |
| $(h, r, t, ts)$ | Timestamped fact. |
| $\mathcal{N}_e$ | Neighborhood of entity $e$. |
| $\mathcal{C}_{(h,r,t,ts)}$ | Context of fact $(h, r, t, ts)$. |
| $\mathbf{h}, \mathbf{r}, \mathbf{t}$ | Embeddings of $h, r$ and $t$. |
| $\mathcal{T}_i$ | Continual learning task $i$. |
| $\mathcal{R}_p$ | Persistent relations in instance-IL setting. |
| $\mathcal{E}_i, \mathcal{R}_i, \mathcal{TS}_i, \mathcal{F}_i$ | Entity, relation, timestamp and fact sets of $\mathcal{G}_i$. |
| $\mathcal{D}_i^{train}, \mathcal{D}_i^{val}, \mathcal{D}_i^{test}$ | Training, validation and test sets of $\mathcal{T}_i$. |

## B  Experimental setting

On both datasets, the embedding dimensions of entity and relation are set to 100. We set the number of self-attention head to 1 and apply drop path to avoid overfitting with a drop rate of 0.2. The maximum number of neighbors for a given entity is set to 50. The number of negative context for each fact is set to 1. The margin $\gamma$ in Eq. 2 is set to 5. For all experiments except for the ablation study on the trade-off parameter, $\beta_0$ and $\beta_1$ in Eq. 11 are set to 20 and 30 respectively. The number of replayed facts is set to 10 except for the ablation study. Trained model is applied on validation tasks each 5 epochs, and the current model parameters and corresponding performance are recorded, after stopping, the model that has the best performance on Hits@10 is treated as final model. For number of training epoch, we use early stopping with 30 patient epochs, which means that we stop the training when the performance on Hits@10 drops 30 times continuously. During training, we apply mini-batch gradient descent to train the model with a batch size of 128 for both datasets. Adam optimizer (Kingma & Ba, 2015) is used with a learning rate of $1e - 4$. All models are implemented by PyTorch and trained on single V100 GPU.

## C  Experimental Results

Figure. 3 and Figure. 4 illustrate the performance of BER on task against baseline methods after training on all six tasks on ICEWS05-15-classIL and ICEWS05-15-expansive, respectilvey. The lighter ones correspond to the testing performance immediately after training on each respective task, while the darker results indicate the final performance after training on Task 6. In general, BER achieves the best overall performance in terms of MRR and Hits@10 on both datasets and strikes a balance between learning new knowledge and maintaining knowledge learned from previous tasks.

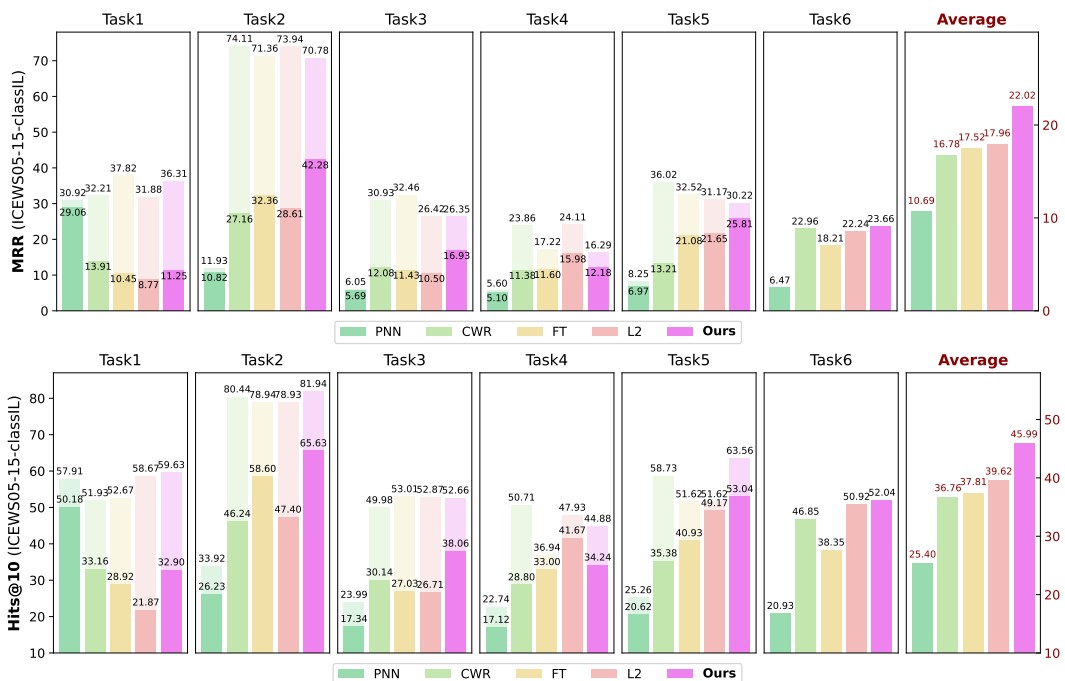

Figure 3: Comparison between different methods of MRR and Hits@10 on ICEWS05-15-classIL. We present the results spanning from Task 1 to Task 6. The lighter ones correspond to the testing performance immediately after training on each respective task, while the darker results indicate the final performance after training on Task 6 (with some amount of forgetting excluding Task 6).

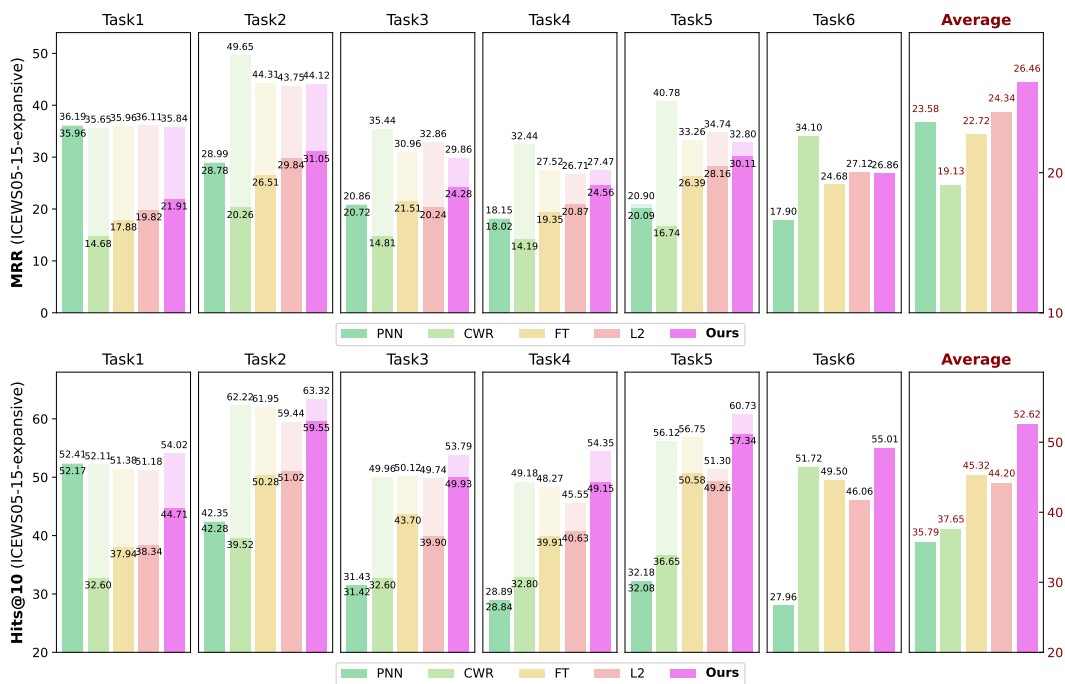

Figure 4: Comparison between different methods of MRR and Hits@10 on ICEWS05-15-expansive.

