# OpenReview forum: "Continual Knowledge Graph Link Prediction: Beyond Experience Replay"
_ICLR.cc/2024/Conference — ICLR 2024 Conference Withdrawn Submission_

### Official Review · Reviewer_kHDH · 2023-10-15

**Soundness:** 3 good
**Presentation:** 3 good
**Contribution:** 3 good
**Rating:** 5
**Confidence:** 4

**Summary:**

This article proposes a new task, continuous knowledge graph link prediction, defines two different task settings, and creates corresponding datasets. The authors introduce a new method called BER for handling this task, solving the catastrophic forgetting problem through experience replay and knowledge distillation.

**Strengths:**

This article is somewhat innovative. It proposes a new continuous knowledge graph (KG) link prediction task. The authors proposed the BER method, which combines experience replay and knowledge distillation and aims to effectively solve the knowledge graph link prediction task. catastrophic forgetting problem.

**Weaknesses:**

In Section 3, although the authors define the continuous knowledge graph link prediction task, as written in the article "continual KG link prediction considers a sequence of tasks", he does not clearly state the missing facts that need to be predicted in each task. As a reader, I feel like I need to understand what this task is about, so there are flaws here. I suggest that the author add an explicit objective function or evaluation metric to the definition.

In Section 4, the authors introduced two new data sets. The article mentions "We partition the ICEWS05-15 dataset into six sequential tasks", dividing the data set into 6 tasks, but does not give each task. The specific distribution of relationships and facts. I suggest that the author provide a table or bar chart in the appendix to show which relationships are included in each task and how many facts each relationship corresponds to.

In Section 5, the authors proposed the BER method but did not give the dimensions and initialization method of the embedding space. In this part of the content, the content of Figure 1 "General framework of BER" was not understood, and no figure was found in the article (Maybe the one in Page 6 is?). For the specific explanation of 1, there are mostly formulas in this section. If you can add some points, I suggest the author add these details in the appendix and explain whether the embedding is normalized or regularized.

In Section 6, although the author has done many experiments and has many results, a more detailed analysis and explanation of these results may also help readers better understand the effectiveness and stability of the method.

**Questions:**

The authors chosed ICEWS05-15 as the benchmark data set in the data set construction, but are there other alternative data sets, or what is the reason why the author chose ICEWS05-15?
The article does not mention directions for future research. Is it possible to discuss future work or possible extensions in this area?

---

### Official Review · Reviewer_JHqC · 2023-10-29

**Soundness:** 2 fair
**Presentation:** 3 good
**Contribution:** 1 poor
**Rating:** 3
**Confidence:** 5

**Summary:**

This paper investigates the task of continual knowledge graph link prediction. It constructs two datasets to simulate two scenarios of relation growth. The paper proposes a method for addressing the continuous link prediction task, which includes a memory sample selection module, a contrastive learning module, and a distillation module. Experimental results show that the proposed method outperforms some baseline methods.

**Strengths:**

1. Well-written and easy to understand.

2. Provide code and datasets, ensuring good reproducibility.

**Weaknesses:**

1. The task of continual knowledge graph link prediction has already been introduced in [1]. The task in [1] includes new relations, entities, and facts. The task proposed in this paper can be considered a subtask of the task presented in [1]. LKGE [1] constructed four datasets to simulate growth in entities, relationships, and facts, and the two datasets created in this paper appear to be related to the "RELATION" and "HYBRID" datasets among them. However, this paper does not cite, discuss, or compare with related work like [1].

2. Embedding-based models differ from other class-incremental tasks, as each emerging entity and relation can be considered a growing class that needs to be learned. However, this paper does not delve into the learning of emerging entities.

3. The proposed method lacks novelty. The innovation of the replay and distillation modules is limited.

4. Only a few simple classical methods are listed as comparison methods. Due to the absence of advanced comparison methods, the performance of the proposed method's advancement is questionable.

5. In knowledge graph representation learning, replay-based methods can be time-consuming. However, this paper does not report on model efficiency. If efficiency is not considered, retraining a new model with all the data seems to be a better approach.

[1] "Lifelong embedding learning and transfer for growing knowledge graphs." AAAI 2023.

**Questions:**

Please see Weaknesses 1, 2, and 5.

---

### Official Review · Reviewer_iVed · 2023-10-30

**Soundness:** 2 fair
**Presentation:** 3 good
**Contribution:** 2 fair
**Rating:** 3
**Confidence:** 4

**Summary:**

The paper studies the continual knowledge graph link prediction problem and proposes to formulate the continual learning task in the scenario of knowledge graphs. Then the authors construct two datasets for the task and propose an attention-based experience replay strategy. Experiments are conducted on the constructed two datasets.

**Strengths:**

1. The paper is well-organized and easy to follow.

2. The paper provides a comprehensive literature review.

**Weaknesses:**

1. The contribution of this paper to the problem formulation is not essential. I do not think formulating the continual learning scenario for KGs is a challenging task. Following the formulation of general continual learning on graphs, we can easily define the problem on KGs.

2. The dataset construction is not from scratch. The authors reorganize and split the existing datasets. It is not difficult to construct a dataset if following the standard setting of continual learning. Therefore, the contribution is not significant.

3. The proposed methodology follows the existing work and is less novel.

**Questions:**

1. The authors claim to employ a contrastive learning-based method to learn expressive entity/relation embeddings. However, for knowledge graphs, there are several criteria to evaluate if the learned embeddings are expressive. For example, if the learned embedding can handle different types of relations. I did not find the corresponding experiments in the paper.

2. The paper does not clearly explain why the proposed method is beyond experience replay. Based on my understanding, the method in this paper is still following the design of experience replay.

---

### Official Review · Reviewer_pgai · 2023-10-30

**Soundness:** 1 poor
**Presentation:** 2 fair
**Contribution:** 2 fair
**Rating:** 3
**Confidence:** 5

**Summary:**

This paper addresses the inadequacy of existing benchmark datasets in evaluating models' abilities for continual KG link prediction due to their reliance on sampling-based methods. A novel approach based on experience replay and knowledge distillation is proposed to alleviate the catastrophic forgetting problem, alongside the introduction of two new benchmark datasets explicitly formulated for continual KG link prediction tasks, encompassing class-incremental and expansive settings. Moreover, extensive evaluation verifies the effectiveness and superiority of the model.

**Strengths:**

1. The authors propose a new knowledge graph embedding method that aims to alleviate the catastrophic forgetting problem.
2. The authors construct two benchmark datasets, ICEWS05-15-classIL and ICEWS05-15-expansive, making some contributions to providing a valid benchmark for the fair evaluation of continual KG link prediction methods.

**Weaknesses:**

1. The paper is not organized clearly, which is not friendly for understanding. For instance, there are lack of discussion about Figure 1.
2. The technical solution of this paper is similar to the reference [1]. Formulas 7, 8, and 9 are similar to formulations 3, 4, and 5 in [1], which It's just for simple use without significant changing.
[1] Hierarchical Relational Learning for Few-Shot Knowledge Graph Completion

**Questions:**

Please refer to the weaknesses.

---

### Author Response · Authors · 2023-11-22
**Response to reviewers**

Dear Reviewers,

After careful consideration, we have decided to withdraw this paper. We genuinely appreciate the constructive criticisms and guidance provided in your reviews. Your expertise and thoughtful comments have given us a clearer perspective on how to enhance and refine this work further.

Best regards,

Paper7309 authors